# Artificial Intelligence in Renal Cell Carcinoma Histopathology: Current Applications and Future Perspectives

**DOI:** 10.3390/diagnostics13132294

**Published:** 2023-07-06

**Authors:** Alfredo Distante, Laura Marandino, Riccardo Bertolo, Alexandre Ingels, Nicola Pavan, Angela Pecoraro, Michele Marchioni, Umberto Carbonara, Selcuk Erdem, Daniele Amparore, Riccardo Campi, Eduard Roussel, Anna Caliò, Zhenjie Wu, Carlotta Palumbo, Leonardo D. Borregales, Peter Mulders, Constantijn H. J. Muselaers

**Affiliations:** 1Department of Urology, Catholic University of the Sacred Heart, 00168 Roma, Italy; 2Department of Urology, Radboud University Medical Center, Geert Grooteplein 10, 6525 GA Nijmegen, The Netherlands; peter.mulders@radboudumc.nl (P.M.); stijn.muselaers@radboudumc.nl (C.H.J.M.); 3Department of Medical Oncology, IRCCS Ospedale San Raffaele, 20132 Milan, Italy; laura.lmarandino@gmail.com; 4Department of Urology, San Carlo Di Nancy Hospital, 00165 Rome, Italy; riccardobertolo@hotmail.it; 5Department of Urology, University Hospital Henri Mondor, APHP (Assistance Publique—Hôpitaux de Paris), 94000 Créteil, France; alexandre.ingels@gmail.com; 6Department of Surgical, Oncological and Oral Sciences, Section of Urology, University of Palermo, 90133 Palermo, Italy; nicpavan@gmail.com; 7Department of Urology, San Luigi Gonzaga Hospital, University of Turin, Orbassano, 10043 Turin, Italy; pecoraro416@gmail.com (A.P.); daniele.amparore@unito.it (D.A.); 8Department of Medical, Oral and Biotechnological Sciences, G. d’Annunzio University of Chieti, 66100 Chieti, Italy; mic.marchioni@gmail.com; 9Andrology and Kidney Transplantation Unit, Department of Emergency and Organ Transplantation-Urology, University of Bari, 70121 Bari, Italy; u.carbonara@gmail.com; 10Division of Urologic Oncology, Department of Urology, Istanbul University Istanbul Faculty of Medicine, Istanbul 34093, Turkey; erdemselcuk1@gmail.com; 11Urological Robotic Surgery and Renal Transplantation Unit, Careggi Hospital, University of Florence, 50121 Firenze, Italy; riccardo.campi@gmail.com; 12Department of Urology, University Hospitals Leuven, 3000 Leuven, Belgium; eduard.roussel@uzleuven.be; 13Section of Pathology, Department of Diagnostic and Public Health, University of Verona, 37134 Verona, Italy; calioanna@gmail.com; 14Department of Urology, Changhai Hospital, Naval Medical University, Shanghai 200433, China; wuzhenjie17cz@gmail.com; 15Division of Urology, Maggiore della Carità Hospital of Novara, Department of Translational Medicine, University of Eastern Piedmont, 13100 Novara, Italy; palumbo.carlotta@gmail.com; 16Department of Urology, Well Cornell Medicine, New York-Presbyterian Hospital, New York, NY 10032, USA; ldb4001@med.cornell.edu

**Keywords:** artificial intelligence, pathology, renal cell carcinoma, kidney cancer

## Abstract

Renal cell carcinoma (RCC) is characterized by its diverse histopathological features, which pose possible challenges to accurate diagnosis and prognosis. A comprehensive literature review was conducted to explore recent advancements in the field of artificial intelligence (AI) in RCC pathology. The aim of this paper is to assess whether these advancements hold promise in improving the precision, efficiency, and objectivity of histopathological analysis for RCC, while also reducing costs and interobserver variability and potentially alleviating the labor and time burden experienced by pathologists. The reviewed AI-powered approaches demonstrate effective identification and classification abilities regarding several histopathological features associated with RCC, facilitating accurate diagnosis, grading, and prognosis prediction and enabling precise and reliable assessments. Nevertheless, implementing AI in renal cell carcinoma generates challenges concerning standardization, generalizability, benchmarking performance, and integration of data into clinical workflows. Developing methodologies that enable pathologists to interpret AI decisions accurately is imperative. Moreover, establishing more robust and standardized validation workflows is crucial to instill confidence in AI-powered systems’ outcomes. These efforts are vital for advancing current state-of-the-art practices and enhancing patient care in the future.

## 1. Introduction

Renal cell carcinoma (RCC) is among the top 10 most common cancers in both men and women. The incidence of RCC has gradually risen in recent years, resulting in increased time-, effort-, and cost-related demands on healthcare systems [1]. Adequate RCC diagnosis and treatment planning relies on adequate clinical data, imaging, histology, and molecular profiling [2,3].

Histological analysis, which is supported by genetic and cytogenetic analysis, is crucial for RCC diagnosis, as well as subtyping and defining features with high prognostic and therapeutic impact [4,5]. These features include tumor grade, RCC subtype, lymphovascular invasion, tumor necrosis, sarcomatoid dedifferentiation, etc. [6,7,8]. RCC histological diagnosis and classification, in particular, can be a daunting task, as it encompasses a broad spectrum of histopathological entities, which have recently been subject to changes [9,10].

Over the years, the daily clinical practice of treating patients with RCC has changed from using paper charts, analog radiographs, and light microscopes to using more modern counterparts, such as electronic health records and digitalized radiology and virtual pathology. This shift has generated an enormous amount of digital data, which can be utilized in data-characterization algorithms or artificial intelligence (AI) [11,12].

The use of AI in radiology, which is also known as radiomics, has shown excellent diagnostic accuracy for detecting RCC and can even provide information regarding RCC subtyping, nuclear grade prediction, gene mutations, and gene expression-based molecular signatures [13]. In line with AI in radiology, efforts to use AI in RCC histopathology have been undertaken in recent years. This relatively new field, which is called pathomics or computational pathology, can be used to improve efficiency, accessibility, cost-effectiveness, and time consumption, as well as enhance accuracy and reproducibility with lower subjectivity [11,14,15,16,17]. In addition, Whole Slide Imaging (WSI) technology allows machine learning in pathology by providing an enormous amount of high-quality information for training and testing AI models to identify specific features and patterns that can be complex for even the human eye to discern [12,18,19]. Ultimately, AI aims to assist pathologists in making more accurate and consistent diagnoses in shorter periods of time and is a valuable implement to undercover the above-cited information [20,21].

In this literature review, we aim to provide an overview of the current evidence regarding the use of computational pathology in histopathology in RCC. Our review aims to evaluate the potential prospects for implementing this emerging technology in everyday practice by comparing and analyzing its advantages and possible drawbacks, as well as bottlenecks that may hinder its development. Furthermore, we explore how this intriguing new technology can aid pathologists in making their work less time consuming, more standardized, and cost effective

## 2. Evidence Acquisition

We conducted a narrative review of the literature concerning all possible applications of AI in the histo-pathological analysis of RCC specimens.

The Medline database was screened, and literature research was restricted to articles published in English between 1 January 2017, and 1 January 2023, since most of the relevant literature in this field was published in this timeframe.

We used a structured search strategy (Appendix A), obtaining 98 results that were reviewed, and references to the retrieved articles were hand-searched to identify additional reports that met the scope of this review.

Original studies and case series were selected for inclusion, while reviews, editorials, and letters to editors were excluded. Finally, references to the retrieved articles were hand-searched to identify additional reports that met the scope of this review.

The titles and abstracts of all papers included were independently assessed against the inclusion and exclusion criteria using Rayyan (Rayyan Systems, Cambridge, MA, USA).

## 3. Basics of Artificial Intelligence and Its Application in Histopathology

Machine learning (ML) is a subfield of AI that uses algorithms that enable computers to learn from digital images of tissue samples. In histo-pathology, it can be used for many tasks, such as digital analysis of images of tissue samples, identification of different structures or cell types, and classification or segmentation of different regions in the tissue sample [22]. The capabilities of ML increased with the development of deep learning (DL), with a section of ML now focused on creating a virtual neural network with multiple layers inspired by the ways through which biological neurons communicate [23]. DL models are well-suited for feature extraction and learning from data because they can automatically identify complex patterns and relationships within large and diverse datasets, such as those used in cancer diagnostics (Definition in Box 1).

Choosing the best algorithm for AI applications in histo-pathology is still challenging. There are three primary types of learning: supervised learning, which uses labeled data for training; unsupervised learning, which finds patterns without labels; and weakly supervised learning, which strikes a medium ground through use of partially labeled data.

In histo-pathological practice, there are numerous time-consuming and repetitive tasks, such analysis of high-volume biopsy tissue samples from breast, prostate, colon, and cervix due to screening programs, as well as finding a large quantity of resected lymphnodes during routine surgical procedures. AI has the potential to flag suspicious regions for inspection and may eventually enable autonomous case assessment.

In addition, AI can help pathologists to complete classification tasks, like highlighting regions of prostate cancer using different colors to represent different Gleason grades [24,25].

Moreover, combining segmentation, detection, and classification techniques makes it possible to objectively quantify established biomarkers utilized in clinical practice. Specific instances are the evaluation of tumor-infiltrating lymphocytes [26] and the quantification of programmed death-ligand 1 (PD-L1)-positive cells [27], which can even be predicted directly via slides [28].

Therefore, AI can be utilized for tasks such as tumor detection and classification, including subtyping, image segmentation, tumor grading, and predictive/prognostic modeling, within the field of histopathology.

Box 1Definitions.**Machine learning:**Machine learning is a specific branch of artificial intelligence, based on algorithms that enable computer systems to learn, make predictions, and decisions based on data, without the need for explicit programming instructions to do so.**Whole-slide images:**Digital representations of entire microscope slides created by scanning glass slides with high-resolution scanners.**Deep learning**:A subfield of machine learning where algorithms are trained for a task or set of tasks by subjecting a multi-layered artificial neural network to a training data. It eliminates the need for manual feature engineering by allowing the networks to learn directly from raw input data during the training process. The acquired algorithm is subsequently utilized for tasks such as classification, detection, or segmentation. The term "deep" refers to the use of artificial neural networks comprising numerous layers, thus referred to as deep neural networks.**Convolutional neural network:**In deep learning, a class of artificial nural network consisting of convolutional of a sequence of convolutional layers to process an input data and produce an output. Each layer implements the convolution operation between the input data and a set of filters. These filter values are learned automatically during training, allowing the network to extract relevant features from the data in an end-to-end fashion (learning the optimal value of all parameters of themodel simultaneously rather than sequentially)**Digital pathology**:The process of digitizing the conventional diagnostic approach. It is accomplished through the utilization of whole-slide scanners and computer screens**Pathomics:**The analysis by computational algorithms of digital pathology data, to extract meaningful features. These features are then used to build models for diagnostics, prognostics, and therapeutics purposes**Computational pathology:**Computational analysis of digital images acquired by scanning pathology slides**Image segmentation:**The process of dividing a digital pathology image into distinct regions or objects of interest (for example nuclei or tumor region) to enable analysis and extraction of specific features.

## 4. Artificial Intelligence Aided Diagnosis of RCC Subtypes

Although several advances have been made in RCC diagnostics in the last decade, especially in imaging techniques, histo-pathological diagnosis based on a pathologist’s skill and experience remains the standard clinical practice used to distinguish RCC from normal renal tissue at the microscopic level [13,29,30,31].

However, RCCs can have complicated characteristics that make the diagnosis difficult, laborious, and time consuming, even for experienced pathologists. These issues are known to lead to a moderate inter-reader agreement for the RCC subtype [32,33,34]. In addition, several studies demonstrated that computational pathology could be a solution to more uniform specimen readings and reduce intra- and inter-observer variability [35,36,37].

### 4.1. RCC Diagnosis and Subtyping in Biopsy Specimens

RCC varies in its biological behavior, ranging from indolent to aggressive tumors. Currently, no reliable predictive models that distinguish between different clinical types are available for use in the pre-operative setting, creating concerns about under- and over-treatment, especially in small renal masses (SRMs), which now represent up to 50% of renal lesions [38,39,40,41,42]. Therefore, this issue can lead to overdiagnosis and overtreatment. To date, there are no highly reliable biomarkers or imaging methods that can correctly differentiate between benign and malignant lesions [43,44,45] As a result, there has been a growing trend of using renal mass biopsy (RMB) to address this challenge over the past decade [46,47,48].

However, RMBs have some limitations as they are non-diagnostic in approximately 10–15% of the cases and remain intrinsically invasive [49]. The main reason for the high percentage of non-diagnostic results is inadequate sampling of tumors [50]. Another crucial issue in RMB is a fair degree of interobserver variability [51], a concern that is also found in breast, prostate, and melanoma biopsies [52,53,54].

To tackle these problems, Fenstermaker et al. developed a DL-based algorithm for RCC diagnosis, grading, and subtype assessment [55]. Their method reached a high accuracy level when using only a 100 square micrometers (µm^2^) patch, making it a potentially valuable tool in RMB analysis. In addition, although their method was trained on whole-mount surgical specimens, a computational method trained and tested on small tissue samples may reduce the need for repeat biopsies by decreasing insufficient tissue sampling and reducing interobserver variability.

However, this study focused on identifying the three main subtypes of RCC without considering benign tumors or oncocytomas. A significant proportion of small renal masses (SRMs) are benign, with oncocytoma being the most frequent benign contrast-enhancing renal mass found. A well-known problem faced by pathologists is differentiating oncocytomas from chromophobe RCC [56,57,58]. Zhu et al. reported favorable results in RCC subtyping in surgical resection and RMB specimens, as well as promising results in oncocytoma diagnosis in RMB [59]. The group trained and tested a model on an internal dataset of renal resections. In addition, they tested this model on 79 RCC biopsy slides, 24 of which were diagnosed as renal oncocytoma, and an external dataset, achieving good performance, as shown in Table 1.

### 4.2. RCC Diagnosis and Subtyping in Surgical Resection Specimens

Despite the recent increased use of RMB and enormous advances in diagnostic accuracy [60,61], approximately 73% of surveyed urologists would not perform a RMB for various reasons [62]. Currently, the standard of treatment for non-metastatic RCC is surgical resection, carried out via either a radical or partial nephrectomy; this technique was also used in some selected cases of metastatic RCC [63,64]. However, examining and analyzing the complex histological patterns of RCC surgical resection specimens under a microscope can be challenging and time consuming for pathologists for many reasons. For instance, nephrectomy specimens exhibit substantial heterogeneity, exemplifying the wide variation observed within RCC surgical resection samples [65]. Moreover, variability among different observers, and even within the same observer, has been reported [33].

Good results were obtained by Tabibu et al. in terms of distinguishing between ccRCC and chRCC and normal tissue using two pre-trained convolutional neural networks (CNN) and replacing the final layers with two output layers, which were fine-tuned using RCC data [66]. Moreover, for subtype classification, the group introduced a so-called directed acyclic graph support vector machine (DAG-SVM) on top of the deep network, obtaining good accuracy in this task. Unlike Tabibu et al.’s model, Chen et al. developed a DL algorithm to detect RCC that was externally validated on an independent dataset [67]. To accomplish this task, they used LASSO (least absolute shrinkage and selection operator), which is a method used in ML to select from a more extensive set of features, i.e., the most important in predicting outcomes. Through LASSO analysis, they identified various image features based on the “The Cancer Genome Atlas” (TCGA) cohort to distinguish between ccRCC and normal renal parenchyma, as well as ccRCC and pRCC and chRCC, obtaining high accuracy in test and external validation cohorts.

Also, Marostica et al. created a pipeline using transfer learning to identify cancerous regions from slide images and classify the three major subtypes, obtaining good performance in both the test set and two external independent datasets (Table 3) [68].

RCC classification is a challenging task not only due to the complexity of the procedure itself, but also because the classification system is subject to periodic updates [69,70]. For example, only in recent years has clear cell papillary renal cell carcinoma (ccpRCC) been recognized as a specific entity [4]. This subtype of RCC histologically resembles both ccRCC and pRCC, and it has clear cell changes. However, ccpRCC has distinct immuno-histochemical and genetic profiles compared to ccRCC and pRCC [71]. It also carries a favorable prognosis relative to the latter carcinoma; therefore, the World Health Organization recently changed its denomination to a clear cell papillary renal cell tumor [72]. Abdeltawab et al. developed a computational model that could distinguish between ccRCC and ccpRCC, obtaining an accuracy of 91% in identifying ccpRCC using the institution files and 90% in diagnosing ccRCC using an external dataset [73].

**Table 1 diagnostics-13-02294-t001:** Overview of studies of AI models for diagnosis and subtyping.

Group	Aim	Number of Patients	Training Process	Accuracy on the Test Set	External Validation (N of Patients)	Accuracy on the External Validation Cohort	Algorithm
Fenstermaker et al. [55]	(1) RCC diagnosis, (2) subtyping, (3) grading	(1) 15 ccRCC; (2) 15 pRCC; (3) 12 chRCC.	No significant error decrease in 25 epochs in training was recorded. Next, a validation dataset was used. Training was halted when the performance on the validation set ceased to improve.	(1) 99.1%; (2) 97.5%; (3) 98.4%	N.A.	N.A.	CNN: 6 different convolutional layers, 2 layers of 32 filters, 2 layers of 64 filters, and 2 layers of 128 filters.
Zhu et al. [59]	RCC subtyping	(1) 486 SR (30 NT, 27 RO, 38 chRCC, 310 ccRCC, 81 pRCC); (2) 79 RMB (24 RO, 34 ccRCC, 21 pRCC).	The models were trained for 40 epochs. The trained model assigned a confidence score for each patch. Finally, a comparison of the trained models was completed.	(1) 97% on SRS, (2) 97% on RMB	0 RO 109 ChRCC 505 ccRCC 294 pRCC:	95% accuracy (only SRs)	DNN: we tested four versions of ResNet: ResNet-18, ResNet-34, ResNet-50, and ResNet-101. ResNet-18 was selected for the highest average F1-score on the developement set (0.96)
Chen et al. [67]	(1) RCC diagnosis, (2) subtyping, (3) survival prediction	(1) and (2) 362 NT, 362 ccRCC, 128 pRCC, 84 chRCC; (3) 283 ccRCC.	LASSO was used to identify RCC-related digital pathological factors and their coefficients in the training cohort. LASSO–Cox regression was used to identify survival-related digital pathological factors and their coefficients in the training cohort.	(1) 94.5% vs. NT (2) 97% vs. pRCC and chRCC (3) 88.8%, 90.0%, 89.6% in 1–3–5 y DFS	(1) and (2) 150 NP, 150 ccRCC, 52 pRCC, and 84 chRCC; (3) 120ccRCC.	(1) 87.6% vs. NP; (2) 81.4% vs. pRCC and chRCC; (3) 72.0%, 80.9%, 85.9% in 1-, 3-, or 5-year DFS.	Segmentation and feature extraction pipeline via CellProfiler: (1) and (2) LASSO; (2) LASSO–Cox regression analysis
Tabibu et al. [66]	(1) RCC diagnosis; (2) subtyping,	(1) 509 NT; (2) 1027 ccRCC; (3) 303 pRCC; (4) 254 chRCC.	Training was terminated when validation accuracy stabilized for 4–5 epochs. Data augmentation included random patches, vertical flip, rotation, and noise addition. Weighted resampling was used to address class imbalance. Training parameters remained unchanged.	(1) 93.9% ccRCC vs. NP 87.34% chRCC vs. NP (2) 92.16% subtyping	N.A.	N.A.	CNN (Resnet 18 and 34 architecture based); DAG-SVM on top of CNN for subtyping.
Abdeltawab et al. [73]	RCC subtyping	(1) 27 ccRCC; (2) 14 ccpRCC.	Each image was divided into overlapping patches of different sizes for feature recognition at different sizes. Multiple CNNs outperformed a single CNN for learning features at different scales. Patch overlap of 50% for learning from diverse viewpoints.	91% in ccpRCC	10 ccRCC.	90% in ccRCC	Three CNNs were used for small, medium, and large patch sizes. The CNNs shared the same architecture: a series of convolutional layers intervened by max-pooling layers, followed by two fully connected layers. Finally, there was a soft-max layer

ccRCC = clear cell renal cell carcinoma, ccpRCC = clear cell papillary renal cell carcinoma, chRCC = chromophobe renal cell carcinoma, CNN = convolutional neural network, DAG-SVM = directed acyclic graph–support vector machine, DFS = disease-free-survival, DNN = deep neural network, LASSO = least absolute shrinkage and selection operator, N.A. = not applicable, NT = normal tissue, pRCC = papillary renal cell carcinoma, ResNet = residual neural network, RMB = renal mass biopsy, SR = surgical resection.

The abovementioned studies were mainly supervised and highly defined for RCC approaches, making them time consuming to conduct. However, the capability to apply knowledge gained from previous experiences to novel situations is a vital skill among human beings. For example, pathologists can use lessons learned outside of their specific subspecialty because several cancer types exhibit common hallmarks of malignancy, as demonstrated by Faust et al., who tested whether a previously trained AI system developed to recognize brain tumor features could be applied to clusters and analyze RCC specimens in an unsupervised fashion [74]. The results showed that grouping cancer regions from non-neoplastic tissue elements matched expert annotations in multiple randomly selected cases. This result, hypothetically, represents a way to demonstrate that unsupervised ML-based methods, which were built for the diagnosis of other cancers, can also be used to diagnose RCC, reducing development and work time.

## 5. Pathomics in Disease Prognosis

The prognosis for RCC depends on several factors, including anatomical and clinical factors, while histological and molecular factors play important prognostic roles in both non-metastatic disease and mRCC [75].

### 5.1. Cancer Grading

Tumor grading is considered to be one of the most critical factors in prognosis prediction, as the 5-year survival rate for patients with low-grade RCC is around 90%, while in high-grade RCC, the survival rate is about 12% [75,76,77].

Although largely replaced by the WHO/ISUP grading classification method, the Fuhrman grading system still acts as an independent factor in determining a higher risk of recurrence and a lower chance of survival [78,79,80,81,82]. The Fuhrman grading system predominantly focuses on the morphology of the nucleus (size and shape) and the existence of prominent nucleoli, though inter- and intra-observer variability is a serious issue [33,37,83]. Yeh et al. trained a support vector machine (SVM) classifier that performed effectively in identifying, size-estimating, and calculating spatial distribution, as well as distinguishing between low and high grades on ccRCC specimens [84]. However, it could not differentiate between specific grades (e.g., III and IV), and no analyses of patients’ likelihood of survival were presented.

Unlike the Fuhrman grading system, the WHO/ISUP system relies solely on nucleolar prominence for grade 1–3 tumors, allowing lower inter-observer variation [85]. Therefore, Holdbrook et al. developed a model that detected prominent nucleoli and quantified nuclear pleomorphic patterns by concatenating features (i.e., combining different features (or variables) into a single input representation for the model) extracted from prominent nucleoli and classifying them as either high- or low-grade features [86]. The model also showed excellent grade classification accuracy and prognosis prediction by comparing these results to a multigene score.

The aforementioned computational systems have many unique features, like image processing, feature extraction, classification method, and predicting two-tiered grades (which demonstrated effective performance in cancer-specific-survival (CSS) prediction). [87]. Tian et al. used 395 ccRCC cases from the TCGA dataset reviewed by a pathologist and stratified via the two-tiered system: low- or high-grade features [88]. Of these features, 277 had concordance between the TCGA and the pathologist’s assigned grade and were used to train the model by extracting different histomic features for each patch. They used LASSO regression to select the features most associated with different grades, obtaining a model that predicted two-tiered ccRCC grading in good agreement with manual grades. It also showed a significant association between the predicted grade and overall survival, even when adjusting for age and gender. Furthermore, the model’s predicted grade was superior in terms of overall survival prediction to TCGA and pathologist grade in discordant cases. This study was different from those of Yeh et al. [84], who only evaluated one feature (i.e., maximum nuclei size) to predict the two-tiered grade, and Holdbrook et al. [86], who used up to four concatenate feature vectors to calculate F-scores before classifying features into low or high grade. The features used in the model of Holdbrook et al. [86] are unspecified.

In addition, Tian et al. and Holdbrook et al. showed that the predicted grade had prognostic value, whereas Yeh et al. did not report any association between their grade and prognosis.

Tian et al.’s study used a conventional image analysis technique for nuclei segmentation. However, DL-based techniques for nuclei segmentation might be viable solutions, as shown by the methods of Yeh et al. and Song et al., to this task [84,89]. The results of the studies mentioned above are summarized in Table 2.

### 5.2. Molecular-Morphological Connections and AI-Based Therapy Response Prediction

Recent developments in predicting RCC survival suggest that molecular differences within subtypes affect prognosis, as well as potentially predictive molecular biomarkers and marker signatures, even though there is no definitive evidence to date supporting the routine clinical use of biomarkers for treatment selection in metastatic RCC (mRCC) [90,91,92,93,94,95].

As the finding of predictive biomarkers still represents an unmet clinical need, AI can be used to explore connections between molecular biomarkers and morphological features on histopathology images, thus overcoming traditional biomarker analysis limitations, such as the high cost (both financially and in terms of time), limited sample size, and lack of standardization [96,97,98,99].

Among the many possible genetic aberrations in RCC, one crucial type of mutation are copy number alterations (CNAs), which are associated with an RCC’s development, treatment response, and prognosis [100,101]. Marostica et al. used transfer learning to develop CNAs and somatic mutation image-based prediction models. They demonstrated that CNAs in several genes, including KRAS, EGFR, and VHL, could affect quantitative histopathology patterns [68]. Furthermore, the group leveraged a framework to predict ccRCC tumor mutational burden, which is a potential yet controversial biomarker for immune checkpoint blockade response [102], and obtained good performances on this task. It is important to note that this approach was weakly supervised and did not need a slide-level label with detailed region or pixel-level segmentation, making it readily applicable for clinical use.

Although immunotherapy has changed the field of mRCC over the last years, TKI monotherapy still plays an essential role in treating patients who are unable to receive or tolerate checkpoint inhibitors as a later-line therapy [75,103]. Go et al. developed an ML-based method to identify which mRCC patients will respond to VEGFR-TKI treatment by analyzing clinical, pathology, and molecular data from 101 patients [104]. Specimens of the primarily resected tissue were collected and retrospectively divided into clinical and non-clinical benefit groups. The authors developed a predictive classifier and obtained a prediction accuracy of 0.87.

As stated, gene expression signatures are commonly used as predictive biomarkers. Endothelial cells and vascular architecture are known to play roles in the biological behavior of the tumor [105]. Ing et al. used ML to analyze tumor vasculature to gather prognostic insights [106]. They used ccRCC cases from the TCGA database to train their algorithm and discovered that nine vascular features correlated with clinical outcomes. They found that four of these features had more significant variation in individuals with poor outcomes than favorable outcomes, linking variation in vascular structure to worse results. Ing et al. identified 14 genes that correlated strongly with these features and built 2 ML-based models with satisfactory prediction outcomes comparable to those of traditional gene signatures. Further efforts are needed to develop models using morphologic and genomic biomarkers to improve patients’ prognosis and treatment options.

Another active area of RCC research is the field of epigenetics [107,108,109,110,111]. Zheng et al. investigated possible interactions between histopathologic features and epigenetic changes in RCC [112]. Using morphometric features extracted from histopathological images, they employed ML models to accurately forecast differential methylation values for specific genes or gene clusters. Furthermore, prospective studies are needed to predict the mechanisms underlying cancer progression using predicted genes [113]. The results of the studies mentioned above are summarized in Table 3.

### 5.3. Prognosis Prediction Models Based on Computational Pathology

In the past, several models were developed and externally validated for the prediction of the prognosis of RCC patients. These models, which are currently used for both localized and metastatic RCC, are mainly based on clinicopathological data, both for localized and mRCC cases [114,115,116,117]. Currently, the prognostic models of localized ccRCC mainly include the Leibovich score [116] and the UISS score [117]. The latter score is primarily based on clinicopathological data, making the pathologist’s subjective experience a limitation of their performances [118,119]. All mentioned models incorporate clinical parameters within their framework; however, models based exclusively on pathological data have been validated [120], Regarding mRCC, risk groups assigned via the Memorial Sloan Kettering Cancer Center (MSKCC) and the International Metastatic Renal Cell Carcinoma Database Consortium (IMDC) may differ in up to 23% of cases [75]. Although these models have shown reasonably good performance in the past, there is still room for improvement [121]. AI multimodal approaches applied to medical issues can raise accuracy by up to 27.7% compared to a single modality [122]. Specifically, integrating an ML-based algorithm that predicts RCC survival from histopathology to other known prognosis modalities improved prediction accuracy in multiple studies [123,124].

Cheng et al. was the first study to combine features from the gene data and histopathologic data for ccRCC prognosis [125], thus generating a risk index strongly correlated with survival and outperforming predictions based on separate consideration of morphologic features or eigengenes. The predicted risk could also stratify early-stage patients (stage I and II), whereas no significant difference in survival outcomes when using stage alone was recorded. In Cheng et al.’s study, microenvironment and radiologic imaging information were not integrated into the prognostic model. At the same time, the latter feature proved to be the single modality with the best predictive performance in a computational method presented by Ning et al. This method combined features extracted from CT, histopathological images, and clinical and genomic data [126]. However, Ning et al.’s method also had limitations, such as a small sample size and a lack of external validation. Another algorithm used by Chen et al. was trained on ccRCC images from the TCGA cohort and validated on Shangai General Hospital images to identify substantial survival-related digital pathological factors and combine them with clinico-pathological factors (age, stage, and grade) [67]. The integration nomogram developed in that study showed good ability in predicting 1- 3- and 5-year DFS (Table 1). The study also defined the cut-off value for high- and low-risk scores as the median score for each cohort. Therefore, external validation using a larger cohort or a prospective study would be necessary to confirm the novel computational recognition model’s validity and determine the optimal cut-off value for high- and low-risk scores.

Another study by Schulz et al. reported on a multimodal deep learning model trained on multiscale histopathological images, CT/MRI scans, and genomic data from whole exome sequencing [127]. The model showed excellent performance in terms of 5-year survival status prediction, as it outperformed other parameters (T-stage, N-stage, M-stage, and grading). They also investigated the possibility of predicting the 5-year survival status by obtaining a significant difference in the survival curves after dividing the cohorts into low- and high-risk patients, even after evaluating only M0 or M + patients. However, this study had the following limitations: it needed to compare other clinical tools that consider factors such as performance status and calcium levels incorporated in the current, which are widely used prognostic models; the external validation sample size was relatively small; and further research is required to confirm the generalizability of the authors’ approach.

The above-mentioned and future models should be externally validated, used in prospective cohorts, and compared to current prognostic models regarding discrimination, calibration, and net benefit [75]. The results of the studies mentioned above are summarized in Table 4.

## 6. Future Perspectives

According to currently available data, AI and ML in RCC pathology (‘pathomics’) hold promise for the future, as they might help us to overcome several problems in classic histopathology, such as intra- and inter-observer variability and time consumption. Currently, several AI methods can be reliable in RCC diagnosis and, on some occasions, appear capable of predicting clinical outcomes in a few seconds. This capability could be of great help for pathologists in times in which the incidence of RCC is still rising. However, this exciting field is still relatively new and not without teething troubles, both in general and specifically within the realm of RCC [128,129].

In this review, we reported on the excellent results achieved using AI in several tasks, like staging and grading. Supervised learning methods efficiently perform these tasks but cannot be visually authenticated. In simple terms, the machine generates an answer (i.e., low or high grade or subtype) according to its learned algorithms, which humans cannot survey. These algorithms are often referred to as black box algorithms [130]. This problem makes them prone to doubt by the pathology community, as the pathologist must have faith in the findings before approving and discussing a report in multidisciplinary meetings [131]. One possible solution might be creating tools that bring transparency to non-linear machine learning techniques. For instance, gradient-weighted class activation mapping (grad-CAM) is a tool that can overlay images and heatmaps to improve visualization of the cell type or region in which the informative features were expressed [132]. Another possible solution can be “searching and matching”, instead of “classifying” in an unsupervised fashion, which the group of Faust et al. used for RCC diagnosis [74]. With unsupervised learning, computers can search and cluster images with matching features in a dataset without labeling the data, which can be labor-intensive and potentially biased [133]. This method more or less resembles the current workflow, as pathologists often use atlases to compare images found in the specimen to judge if they match certain previously described conditions. Alternatively, asking other experts for a second opinion may be useful. However, this approach does not exclude the intervention of human experts since a pathologist still needs to inspect and interpret the images visually.

Another possible drawback of computational pathology is the current lack of generalization due to potentially biased inputs used in the training processes of models. For example, using cross-validation, ML models are validated using a set different from the training set, which can lead to biased evaluation if the input data are biased. Therefore, a recommended step before model training is to always check for any potential sample bias and assess whether there may be any issues related to sample size [134,135], heterogeneity [136], noise [137], and confounding factors [138].

Moreover, supposing the data are derived from one pathology laboratory, the algorithm may only be able to account for some variations and artifacts arising from different institutions. For example, the color distribution of WSIs varies across different pathology laboratories due to the staining process.

Once the data are adequately processed, the model is trained using the training set, and its performance is evaluated using the validation set. The so-called ‘overfitting’ can occur when a model is so finely tuned to a particular dataset that it fails to generalize well to new and unseen data. Overfitting is akin to memorizing answers to a test rather than understanding the material. Once the training process is complete, the final performance of the model is evaluated using the test set, which contains data that the model has not seen before that moment. This final evaluation estimates the model’s performance using new and unseen data [139]. But, if the model is overfitting, it can still perform well if the data are derived from the same laboratory.

This approach leads to inter-center variability that impacts the accuracy of machine learning algorithms used to automatically analyze WSIs. This issue affects state-of-the-art CNN-based algorithms, which often exhibit reduced performance when applied to images from a different center than that on which they were trained [22,23,140,141]. Therefore, a global standard for tissue processing, staining, slide preparation in surgical pathology, and even digital acquisition would be greatly helpful [142]. Existing solutions to reduce generalization error in this setting can be categorized into stain color augmentation and stain color normalization, with ML-based methods that perform stain color normalization using a neural network being proposed [143]. One of the most effective methods to mitigate overfitting is external validation, which involves testing the method on a group of new patients distinct from the initial set, thus assessing the model’s generalization ability [20].

The critical evidence for generalizability would be introducing external validation. Any features selected based on idiosyncrasies in the original training data, such as technical or sampling biases, would likely not function properly. As a result, adequate performance while using a reasonably extensive external validation set is seen as evidence of a model’s generalizability (Figure 1 and Figure 2) [144].

Additionally, it is important to note that, as stated above, radiomics showed promising results in different tasks, in particular in diagnosing and subtyping tasks. Many studies used histopathology results as the reference standard to evaluate the radiomic model [145]. Over the past decade, computational pathology research experienced a shift in focus. Initially, the aim of research was to replicate the diagnostic process already conducted by pathologists. However, the most recent literature witnessed a move towards uncovering and exploring “sub-visual” prognostic image cues derived from histopathological images.

Radiomics involves the extraction of computational features that quantify tissue heterogeneity at the macroscopic level by leveraging ML. In contrast, pathomics focuses on providing quantitative information at the micro scale. The fusion of radiomics and pathomics can offer, in the future, an opportunity to combine tumor heterogeneity at both the macro and micro scales, potentially enhancing the integrated signature through complementary insights [146].

To conclude, AI is a promising tool that remains under investigation in relation to the diagnosis, grading, prognosis assessment, and treatment of kidney neoplasms. Results of new AI algorithms are encouraging since they are either on par with or outperform current state-of-the-art methods. However, most available technologies are currently unavailable for widespread clinical use, and further evidence is needed regarding their efficacy. Therefore, further advancements in this exciting field are eagerly awaited [23].

## Figures and Tables

**Figure 1 diagnostics-13-02294-f001:**
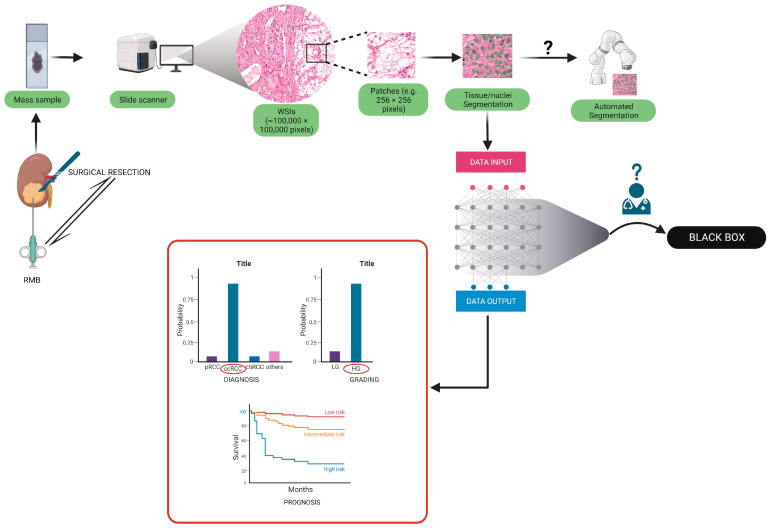
Pathway for the development of pathomics algorithms. After the sample is via by surgical resection or biopsy, the WSI is created and derived patches utilized through a digital scanner to train the algorithm to define diagnostic, prognostic, or predictive models. Supervised learning-based algorithms could carry the “black box” issue (see Section 6).

**Figure 2 diagnostics-13-02294-f002:**
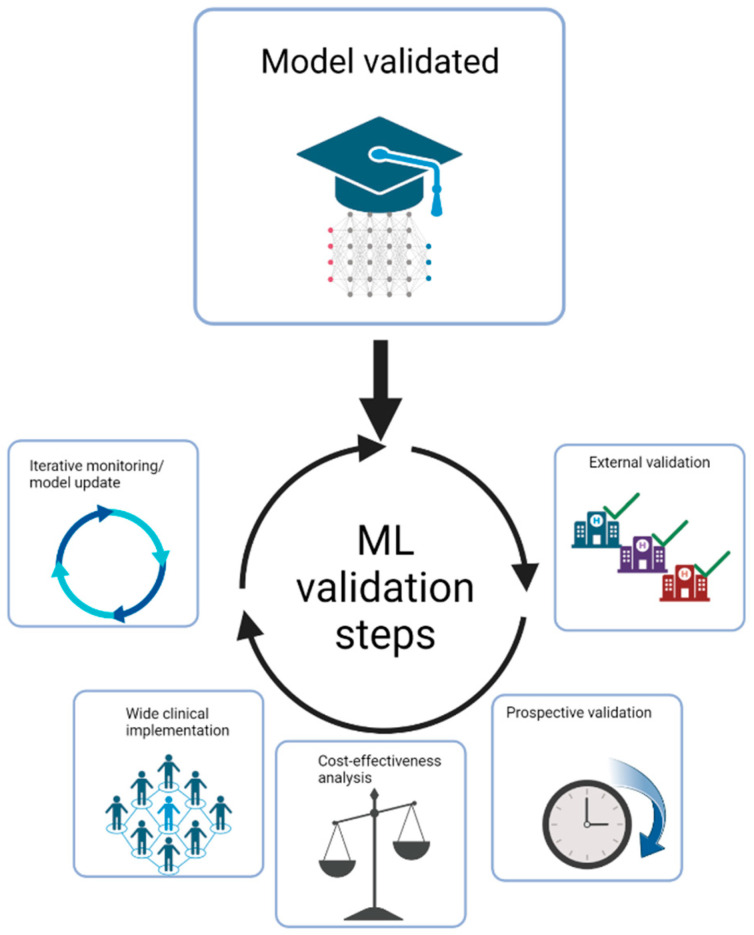
Challenges in clinical translation after the development of a new ML algorithm.

**Table 2 diagnostics-13-02294-t002:** Overview of studies on AI models for RCC grading.

Group	Aim	Number of Patients	Training Process/Methodologies	Accuracy on the Test Set	External Validation (N of Patients)	Accuracy on the External Validation Cohort	Algorithm
Yeh et al. [84]	RCC grading	39 ccRCC	Pixels from the nuclei were manually selected to further train a SVM classifier to recognize nuclei. A person with no special training in pathology engaged in training the classifier with an interactive interface.	AUC: 0.97	N.A	N.A.	WSI analysis with an automatic stain recognition algorithm. An SVM classifier was trained to recognize nuclei. Sizes of the recognized nuclei were estimated, and the spatial distribution of nuclear size was calculated using Kernel regression.
Holdbrook et al. [86]	(1) RCC grading; (2) survival prediction.	59 ccRCC	A cascade detector of prominent nucleoli (constructed by stacking 20 classifiers sequentially) was trained with WSI images to extract image patches for subsequent analysis. This pipeline used two nucleoli detectors to extract prominent nucleoli image patches.	(1) F-score: 0.78–0.83 grade prediction; (2) High degree of correlation (R = 0.59) with a multigene score.	N.A.	N.A.	An automated image classification pipeline was used to detect and analyze prominent nucleoli in WSIs and classify them as either low or high grade. The pipeline employed ML and image pixel intensity-based feature extraction methods for nuclear analysis. Multiple classification systems were used for patch classification (SVM, logistic regression and AdaBoost).
Tian et al. [88]	(1) RCC grading, (2) survival prediction	395 ccRCC	Seven ML classification methods were used to categorize grades based on nuclei histomics features were evaluated. Among these methods, LASSO regression demonstrated the highest performance with a built-in feature selection capability. LASSO regression and its optimal hyper parameter selected the final list of histomics features most associated with grade.	(1) 84.6% sensitivity and 81.3% specificity grade prediction; (2) predicted grade associated with overall survival (HR: 2.05; 95% CI 1.21–3.47).	N.A.	N.A.	Nuclear segmentation occurred, and 72 features were extracted. Features associated with grade were identified via a LASSO model using data from cases with concordancet between TCGA and Pathologist 1. Discordant cases were additionally reviewed by Pathologist 2. Prognostic efficacy of the predicted grades was evaluated using a Cox proportional hazard model in an extended test set created by combining the test set and discordant cases.

AUC = area under curve, ccRCC = clear cell renal cell carcinoma, CI = confidence interval, HR = hazard ratio, DNN = deep neural network, LASSO = least absolute shrinkage and selection operator, N.A. = not applicable, SVM = support vector machine.

**Table 3 diagnostics-13-02294-t003:** Studies aimed to uncover molecular-morphological connections and/or AI-based therapy response prediction.

Group	Aim	Number of Patients	Training Process/Methodologies	Accuracy on the Test Set	External Validation (N of Patients)	Accuracy on the External Validation Cohort	Algorithm
Marostica et al. [68]	(1) RCC diagnosis; (2) RCC subtyping; (3) CNAs identification; (4) RCC survival prediction; (5) Tumor mutation burden prediction.	(1) and (2): 537 ccRCC, 288 pRCC, and 103 chRCC; (3) 528 ccRCC, 288 pRCC, and 66 chRCC; (4) 269 stage I ccRCC; (5) 302 ccRCC.	(1) Weak supervision approach used for malignant region identification; (2) Same transfer learning approach trained for 15 epochs; (3) Independent models for ccRCC, pRCC, and chRCC were developed; (4) 10-fold cross-validation was employed. Upsampling of uncensored data points was performed in each fold’s training set to enhance the model training process.	(1)AUC: 0.990 ccRCC, 1.00 pRCC, 0.9998 chRCC; (2) AUC: 0.953 (3) ccRCC KRAS CNA: AUC = 0.724, pRCC somatic mutations: AUC: 0.419–0.684; (4) Short vs. long-term survivors log-rank test P = 0.02, n = 269; (5) Spearman’s correlation coefficient: 0.419	(1) and (2) 841 ccRCC, 41 pRCC, and 31 chRCC.	(1) 0.964–0.985 ccRCC; (2) 0.782–0.993	(1) Three DCNN architectures (VGG-16, Inception-v3, and ResNet-50) were compared for each task. (2) Same transfer learning approach as above was used. The hyperparameters of DCNNs were optimized via Talos. (3) Two transfer learning approaches were used: gene-specific binary classification and multi-task classification for all genes for CNAs. DCNNs were used for associations between genetic mutations and WSI images. (4) DCNN models used image patches as inputs, predicting binary values for each patient. Grad-CAM was generated to identify the regions of greatest importance for survival prediction.
Go et al. [104]	RCC VEGFR-TKI response classifier; survival prediction.	101 m-ccRCC	ML approaches were applied to establish a predictive classifying model for VEGFR-TKI response. A 10-fold-cross-validated SVM method and decision tree analysis were used for modeling	Apparent accuracy of the model: 87.5%; C-index = 0.7001 for PFS; C-index of 0.6552 for OS	N.A.	N.A.	Features that showed the statistical differences between the good and bad-response groups were selected, and the most appropriate cut-off for each feature was calculated. Secondary feature selection was performed using SVM to develop the most efficient model, i.e., the model showing the highest accuracy with the least number of features
Ing et al. [106]	(1) RCC vascular phenotypes; (2) survival prediction; (3) identification of prognostic gene signature; (4) prediction models.	(1), (2), and (3): 64 ccRCC; (4) 301 ccRCC.	A stochastic backwards feature selection method with 1500 iterations was applied to identify the subset of VF with the highest predictive power. Two GLMNET models were trained: one model was trained on VF-risk groups, and the other model was trained using a 24-month disease-free status as the ground truth for a validation cohort.	(1) AUC = 0.79; (2) log-rank *p* = 0.019, HR = 2.4; (3) Wilcoxon rank-sum test *p* < 0.0511; (4) C-Index: Stage = 0.7, Stage + 14VF = 0.74, Stage + 14GT = 0.74.	N.A.	N.A.	Quantitative analysis of tumor vasculature and developement of a gene signature. The algorithms trained in this framework classified with SVM and random forest classifiers, i.e., endothelial cells, and generated a VAM within a WSI. By quantifying the VAMs, nine VFs were identified, which showed a predictive value for DFS in a discovery cohort. Correlation analysis showed that a 14-gene expression signature related to the 9VF was discovered. The two GLMNET were developed based on these 14 genes, separating independent cohorts into groups with good or poor DFS, which were assessed via Kaplan–Meier plots.
Zheng et al. [112]	RCC methylation profile	326 RCC (also tested on glioma)	In total, 30 sets of training/testing data were generated. Binary classifiers were fitted on the training set, and the best parameters were selected using 5-fold cross-validation. Logistic regression with LASSO regularization, random forest, SVM, Adaboost, Naive Bayes, and a two-layer FCNN were used with optimized parameters.	Average AUC and F1 score higher than 0.6	N.A.	N.A.	To demonstrate that DNA methylation can be predicted based on morphometric features, different classical ML models were tested. Binary classifiers for each task were evaluated using accuracy, precision, recall, F1-score, ROC curve, AUC score, and precision–recall curves. Scores from 30 training/testing data sets were averaged per task. For logistic regression, feature importance analysis was conducted to rank the influence of morphometric features on the prediction task.

AUC = area under curve, ccRCC = clear cell renal cell carcinoma, chRCC = chromophobe renal cell carcinoma, CNA = copy number alteration, DCNN = deep convolutional neural network, DFS = disease-free survival, FCNN = fully connected neural network, GLMNET = elastic-net regularized generalized linear models, Grad-CAM = gradient-weighted class activation mapping, LASSO = least absolute shrinkage and selection operator, ML = machine learning, N.A. = not applicable, OS = overall survival, PFS = progression-free survival, pRCC = papillary renal cell carcinoma, ROC = receiver operating characteristic, SVM = support vector machine, VAM = vascular area mask, VEGFR-TKI = vascular endothelial growth factor receptor–tyrosine kinase inhibitor, VF = vascular features.

**Table 4 diagnostics-13-02294-t004:** Prognostic models.

Group	Aim	Number of Patients	Training Process/Methodologies	Accuracy on the Test Set	External Validation (N of Patients)	Accuracy on the External Validation Cohort	Algorithm
Ning et al. [126]	RCC prognosis prediction	209 ccRCC	The training procedures employed 10-fold cross-validation. Survival distributions of low- and high-risk groups were estimated using the Kaplan–Meier estimator and compared via the log-rank test. The performance of prognostic prediction was assessed using the C-index.	Mean C-index = 0.832 (0.761–0.903)	N.A	N.A.	Two CNNs with identical structures were employed to extract deep features from CT and histopathological images. Histological patches were carefully reviewed by two pathologists to confirm coverage of tumor cells. Global pooling and fully connected layers were utilized at the end of the network to integrate information from all feature maps and make predictions. The BFPS algorithm was employed for feature selection.
Cheng et al. [125]	RCC prognosis prediction	410 ccRCC	A two-level cross-validation strategy was used to validate our method. In the first level, a single patient was chosen as the test set, with the rest used as training sets. The second level was a 10-fold cross-validation performed in the training set to select the best regularization parameter. A regularized Cox proportional hazards model was built on the training set using the selected parameter and based on the model; risk indices of all patients were also calculated.	Log-rank test *p* values < 0.05	N.A.	N.A.	The unsupervised segmentation method for cell nuclei and features extraction was used. lmQCM was used to perform gene coexpression network analysis. The LASSO-Cox model for prognosis prediction calculated the risk index for each patient based on their cellular morphologic features and eigengenes
Schulz et al. [127]	RCC prognosis prediction	248 ccRCC	Unimodal training was conducted. This method was followed by multimodal training, which used the pre-trained weights from unimodal training. Training lasted for 200–400 epochs, and the best model was selected based on the convergence of training and validation curves. The standard Cox loss function was employed for survival analysis, while the cross-entropy loss function was used for binary classification tasks.	A mean C-index of 0.7791 and a mean accuracy of 83.43%. (prognosis prediction)	18 ccRCC	Mean C-index reached 0.799 ± 0.060 with a maximum of 0.8662. The accuracy averaged at 79.17% ± 9.8% with a maximum of 94.44%.	CNN consisting of one individual 18-layer residual network (ResNet) per image modality (histopathology slides, CT scans, MR scans) and a dense layer for genomic data. The network outputs were then combined using an attention layer, which assigned weights to each output based on its relevance to the task at hand. The combined outputs were passed through a fully connected network. Depending on the specific case, either C-index calculation or binary classification for 5YSS was performed. The 5YSS category included patients who either survived for longer than 60 months or passed away within five years of diagnosis.

5-YSS = 5-year survival status, BFPS = block filtering post-pruning search, ccRCC = clear cell renal cell carcinoma, CNN = convolutional neural network, LASSO = least absolute shrinkage and selection operator, lmQCM = local maximum quasi-clique merging, ML = machine learning, N.A. = not applicable, SVM = support vector machine.

## Data Availability

https://www.cancer.gov/ccg/access-data (the cancer genomic atlas program) (accessed on 31 May 2023).

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
