# Peer review of "Artificial Intelligence in Renal Cell Carcinoma Histopathology: Current Applications and Future Perspectives"

_diagnostics, 2023, doi:10.3390/diagnostics13132294_

Round 1

Reviewer 1 Report

I read the article titled with " Artificial intelligence in renal cell carcinoma histopathology: current applications and future perspectives ". The article can be published after the revision. My suggestions are below. Please clarify the main contributions/novelties of this study for readers to better understand this paper.

1.       Why is there no abstract in the article? abstract section should be added.

2.       Graphical abstract should be added.

3.       The contributions are not stated clearly in the Introduction section.

4.        Some abbreviations do not have explanations. The abbreviation table should be attached to the article. (as an appendix)

5.        Recent studies (2022,2023) should be added to the references.

6.       The number of literature in Tables 1,2,3 and 4 should be increased.

7.       The dataset section should be added. Comparison table should be added according to data sets.

8.       The English and presentation can be improved.

 The English and presentation can be improved.

Reviewer 2 Report

This paper review explores the use of computational pathology in histopathology for renal cell carcinoma (RCC). It discusses the shift from traditional methods to digital technologies and highlights the potential of machine learning (ML) and deep learning (DL) in analyzing digital images of tissue samples. The paper emphasizes the benefits of AI in improving diagnostic accuracy and efficiency in RCC diagnosis and subtyping. The use of Whole Slide Imaging (WSI) technology is also mentioned as a valuable tool for training and testing AI models. Overall, the paper aims to enhance the understanding of RCC characteristics and improve patient outcomes through AI-assisted histopathology. The following points should be revised.

1. The introduction section needs to clearly articulate the research background and objectives. The current section contains a lot of general information about renal cell carcinoma (RCC) but does not explicitly state the research objectives and focus of this paper. Please specify the purpose of this paper and provide a concise and clear research question.

2.  Please provide more explanations and context when introducing the concepts of machine learning (ML) and deep learning (DL) to ensure that readers understand these terms. Additionally, consider providing some specific examples to illustrate the applications of ML and DL in histopathological research.

3. The paper mentions the application of AI in radiology but does not explicitly state its relevance to RCC. Please provide more information about the specific applications and advancements of AI in RCC radiology and compare them with computational pathology.

4. The potential advantages of computational pathology in improving efficiency, accuracy, and reproducibility are mentioned but without specific research results or examples to support these points. Please include some relevant studies and cases in the manuscript to support the stated advantages.

5. When discussing the DL-based algorithms and models, provide more details about the specific architectures, training processes, and evaluation metrics used. This will help readers understand the technical aspects of the models and their performance.

6. When discussing the computational models and techniques used in grading, provide more details about the specific algorithms, methodologies, and metrics used for evaluation. This will help readers understand the technical aspects of the models and their performance.

7. More technical details of existing solutions should be discussed.

8. It is better to give some quantitative comparisons of existing solutions.

NA

Round 2

Reviewer 1 Report

I appreciate the effort made by authors in addressing my observations. The manuscript has improved and the authors managed to address my questions. In my view, the paper can be accept.

It should be checked before it is published.

Reviewer 2 Report

All my concerns have been addressed. I recommend this paper for publication.

NA